# Fertimetro, a Principle and Device to Measure Soil Nutrient Availability for Plants by Microbial Degradation Rates on Differently-Spiked Buried Threads

**Giuseppe Concheri**, **Stefano Tiozzo, Piergiorgio Stevanato, Francesco Morari, Antonio Berti, Riccardo Polese, Maurizio Borin and Andrea Squartini \***

Department of Agronomy, Food, Natural Resources, Animals, and Environment DAFNAE, University of Padova, Agripolis, Viale dell'Università 16, 35020 Legnaro (PD), Italy; giuseppe.concheri@unipd.it (G.C.); stefano.tiozzo@unipd.it (S.T.); stevanato@unipd.it (P.S.); francesco.morari@unipd.it (F.M.); antonio.berti@unipd.it (A.B.); riccardo.polese@unipd.it (R.P.); maurizio.borin@unipd.it (M.B.)

\* Correspondence: squart@unipd.it; Tel.: +39-049-827-2923; Fax: +39-049-827-2929

**Abstract:** A novel patented method (PCT/IB2012/001157: Squartini, Concheri, Tiozzo, University of Padova) and the corresponding application devices, suitable to measure soil fertility, are presented. The availability or deficiency of specific nutrients for crops is assessed by monitoring the kinetics of progressive weakening of cotton or silk threads due to in situ microbial activity. The method is based on a nutrient-primed incremented substrate degradation principle. Threads are buried as is or pre-impregnated with N or P solutions, and the acceleration of the degradation rate for the N-supplemented or P-supplemented thread, in comparison to the untreated thread, is proportional to the lack of the corresponding nutrient in that soil. Tests were validated on corn crops in plots receiving increasing fertilizer rates in a historical rotation that has been established since 1962. The measurement carried out in May significantly correlated with the subsequent crop yields recorded in October. The analysis allows an early, inexpensive, fast, and reproducible self-assessment at field level to improve fertilization rates. The device is envisaged as a user-friendly tool for agronomy, horticulture, and any environmental applications where organic matter cycling, soil quality, and specific nutrients excess or deficiency are critical considerations.

**Keywords:** Fertimetro; soil fertility; soil nutrient deficiency assessment; soil fertilization; microbial degradation

## 1. Introduction

Soil microorganisms play a key role in the degradation and recycling of organic matter, governing the activities of mineralization, which allow the release of inorganic nutrients for the benefit of plant roots. For this reason, in microbial ecology a variety of methods have been tested to gather information about their activity in the soil. These include the response to the addition of soluble organic nutrients upon measuring the resulting increased respiration activity by $CO_2$ emission [1], and the quantification of the microbial biomass [2]. The concept of Metabolic Microbial Quotient was later defined [3], while more recently, analyses took advantage of molecular techniques based on qualitative and quantitative analysis of DNA [4], up to omics techniques [5,6]. But these investigations, that often require laborious processing, high costs, and demanding instrumentation, do not necessarily provide a straight and clear interpretation of the situation from an agronomical standpoint. A gap remains; therefore, open

between soil microbial activity and agricultural practice, which urges the development of new methods that could conjugate versatility, cost-effectiveness, and simplicity of use, and which would serve as sensitive indicators of a soil mineralizing potential as well as actual needs for correct fertilization.

The principle hereby presented draws its origin from a classic approach known as cotton strip assay, based on the notion that cellulolytic activities are carried out by widespread soil microbiota and are useful proxies for microbial activity and organic matter turnover. In the early seventies the idea of burying a piece of fabric to inspect its subsequent decay was first adopted in environmental studies focusing on the C cycle in the tundra biome [7]. The use of the test was stimulated by the specification of the British Standard n. 2576 [8] and was then reviewed [9,10]. Other authors tested cellophane as an alternative to cotton [11]. The reaction variability as a function of environmental conditions was also modeled [12]. Subsequently, the principle was applied to the monitoring of soil pollution by heavy metals [13] and aromatic hydrocarbons spills [14]. As alternatives to the tensile strength reduction, methods have been also proposed based on digital imaging of the progressive fabric browning, to be measured by densitometry software [15]. In spite of such a long tradition; however, the cotton strip assay technique suffers from the three following drawbacks that have limited its application in custom agronomical practice, confining it mostly within the research context: (1) Relatively wide cotton bands were used, and their breakage test required machines used in the textile fabric and clothing industry, which are impractical to be proposed as farm-level instrumentation; (2) in-soil incubation times are very long (30–45 days), due to the width of the strips; and (3) the classic method is based solely on the comparison to an unburied piece of the same material, thus it only gives an indication of the overall cellulolytic activity of the soil but no agronomical information, such as the soil content in key plant growth-limiting nutrients like nitrogen and phosphorus.

For these reasons the principle was revisited and a solution to the mentioned drawbacks was devised, leading to a novel concept and method which has been granted patent eligibility (PCT/IB2012/001157) by the European Patent Office.

The key innovative traits of the method are: (a) Thin threads are used instead of large strips, which can; therefore, be broken by applying at least two kilograms of force for the unburied standard. This enables testing of residual strength without any complex equipment. (b) The buried incubation time can be shortened from weeks to days. (c) The method adopts a nutrient-primed incremented substrate degradation principle. Therefore, it can allow for the evaluation of agricultural soil deficiencies and correct these, in a timely manner, to optimize its fertility and crop productivity. The effect of plant nutrients can be evaluated by comparing the degradation of the plain fiber thread with that of threads previously soaked in specific nutrient solutions (N, P, K, oligoelements, or whichever compounds whose biologically-efficient availability one needs to assess in the soil). The dosages of the presoaking baths are formulated to suit the optima of the soil microorganisms that can act on the fiber. This allows measurement of both basal cellulolytic activity and values of the same activities when induced by added N or P. (d) The different fibers include not only cellulose (cotton) but also protein (silk), providing, also, the assessment of proteolytic activities and offering a broader picture of soil organic matter degradation by testing both plant- (cotton) and animal-origin (silk) materials. An automatic version of the Fertimetro has also been assembled, in which the threads are kept already under tension by springs and a datalogger records the time of the breaking event [16].

In the present report the basic version of Fertimetro is introduced and the results of field trials on corn crops, in soils with different productivity levels, are presented. The soils that hosted the tests are part of an historical long-term crop fertilization experiment that was started in the early sixties and that offers plots that have received, for decades, serial dosages of different fertilization regimes.

The working hypothesis was that the Fertimetro methodology would be sensitive enough to capture the existing soil productivity differences and report them as a function of the microbial degradation intensity on the threads during the one-week incubation.

## 2. Materials and Methods

### 2.1. The Experimental Site

The field experiment at the experimental farm "Lucio Toniolo" at the University of Padua includes mono-successional trials of corn that were established in 1962, and that have received different fertilization rates ever since [17–19]. The plowing depth over the years has been decreased from 40–50 cm to 25–30 cm. Organic fertilizers are distributed before plowing, around October, and are added to the slurry residue. Inorganic fertilizers are supplied in April, before sowing the spring crops. Rates of organic and inorganic fertilizers applied are reported in Table 1.

**Table 1.** Fertilization regime yearly-repeated on the corn monoculture plots used in the present trial.

| Designation | Treatment |
|---|---|
| 0 | No fertilization |
| 0 + r | No fertilization + crop residues |
| L1 + M | Manure (30 t ha$^{-1}$ year$^{-1}$) + mineral fertilizer (150 N + 75 P$_2$O$_5$ + 210 K$_2$O kg ha$^{-1}$ year$^{-1}$) |
| L2 | Manure (60 t ha$^{-1}$ year$^{-1}$) |
| Lq1 + M + r | Slurry (60 t ha$^{-1}$ year$^{-1}$) + mineral fertilizer (150 N + 75 P$_2$O$_5$ + 210 K$_2$O kg ha$^{-1}$ year$^{-1}$) + crop residues |
| Lq2 + r | Slurry (120 t ha$^{-1}$ year$^{-1}$) + crop residues |
| M2 | Mineral fertilizer (300 N + 150 P$_2$O$_5$ + 420 K$_2$O kg ha$^{-1}$ year$^{-1}$) |
| M2 + r | Mineral fertilizer (300 N + 150 P$_2$O$_5$ + 420 K$_2$O kg ha$^{-1}$ year$^{-1}$) + crop residues |

The farm is located at the eastern side of the Po River Valley, representing the main agricultural belt in Italy, in the Legnaro area (45°21′ N; 11°58′ E, 6 m a.s.l.). The climate of Legnaro is sub-humid, with average annual values of precipitation equal to 825 mm and a temperature of 12.5 °C.

The local soil, classified as Fluvi-Calcaric Cambisol under the FAO (Food and Agriculture Organization)and UNESCO (United Nations Educational, Scientific and Cultural Organization) parameters, formed and evolved on the alluvium that was deposited by the flooding of the rivers Brenta and Bacchiglione, which created a flood plain that is now part of the drainage basin area of the Venice Lagoon. The soil has a pH 7.53 (KCl); 19.3% clay, 38.3% silt, 42.4% sand, 1.6% organic matter; 0.176% P$_2$O$_5$ (based on Olsen available phosphorus determination) and 18.6% total carbonates; its profile is deep, with no pebbles; and the texture is light-sandy for over a meter of depth, shifting thence, progressively, to sandy-clay [20]. The experimental layout of the whole station is in randomized blocks consisting of 288 plots (96 with three replicates each), with sizes of 7.8 × 6 m. The plots used for the present analysis numbered 24, all cropped with a repeated corn monoculture since 1989, and consistently receiving a given fertilization regime, as reported in Table 1. The installation period of the devices was the last week in the month of May. Agronomical practices adopted are the ones previously described [19].

The scheme includes mineral or organic fertilizers alone or in combination with the option of removing the crop residues or burying them on site. The cattle manure used contained 13.9% C, 0.5% N, 0.2% P$_2$O$_5$. Slurry (liquid bovine livestock waste) had a mean of 4.24% solid matter $\pm$ 0.86 SD; manure (composted excreta with straw) averaged 89.18% solid matter $\pm$ 5.14 SD. Phosphorus content on dry matter in each was 0.61% $\pm$ 0.05 SD and 0.65% $\pm$ 0.07 SD, respectively. The phosphorus amounts delivered by the fertilization scheme are reported in Table 1.

Corn yield was determined using a plot combine harvester by in-field weighing of the grain or straw.

The experimental setup and block randomization design are the ones established in the farm since the beginning of the long-term fertilization trial [19].

*2.2. Analysis of Soil Microbial Activities on Threads*

The extent of microbial degradation of the fibers was assessed using the above-mentioned method, depicted in Figure 1.

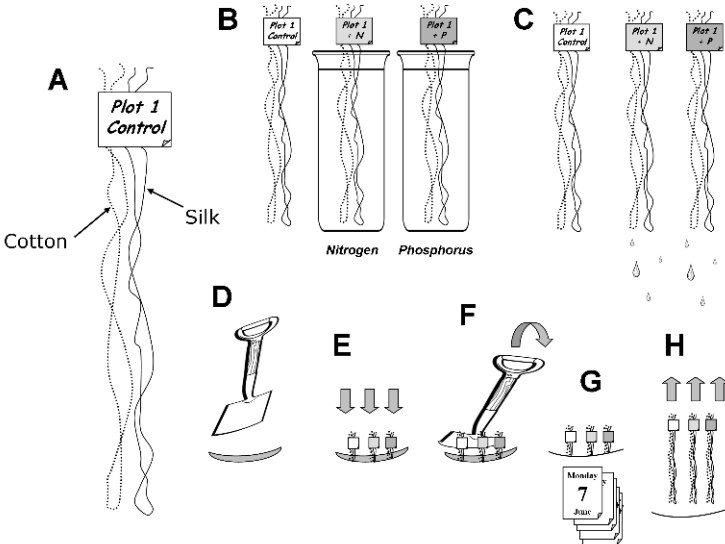

**Figure 1.** Fertimetro methodology. (**A**) cotton and silk filaments folded with a loop and secured with a scotch tape piece; (**B**) bathing of replicated bundles in the solutions of nutrients, whose deficiencies in soil are to be assessed; (**C**) air drying after the bath; (**D**) opening the slot in soil with a spade; (**E**) sliding in the bundles; (**F**) earthing-up, horizontally, from a parallel cut in order for the soil to adhere well to the threads, and closing the slot; (**G**) one-week incubation; (**H**) rescuing of ready-for-measurement bundles.

Filaments were purchased from Cucirini Tre Stelle Manifattura di Settala Spa, Via Pietro Gobetti, 12, 20090, Caleppio di Settala Milan, (http://www.cucirinitrestelle.com). The cotton thread was Nr. 16, while the silk was "Bozzolo Reale" (Royal Cocoon) Nr.24. Both were chosen in white color as their different textures (smoother for silk) allowed for an easy recognition of each. Forty centimeter-long stretches of both types of threads were cut, folded once, and fastened together near the cut extremities by a piece of scotch tape to form a bundle, in which both the cotton and the silk thread featured an end loop. The tape was also used as labels, reporting sample location, date, or other metadata. We prepared three sets for each sampling point, one was untreated while the other two underwent a 15 min soaking in a solution containing either N (3 g/L $NH_4NO_3$) or P (6 g/L $Na_2HPO_4$ + 3 g/L $KH_2PO_4$), whose concentrations can represent optimal values for microbial requirements of N and P [21]. The bundles were then hung to air-dry overnight. For convenience, the scotch tape color of the untreated bundle was white, while those of the bundles bathed in N or P were respectively in red or green.

For the in-soil exposure, a vertical cut was made in the soil with a spade, and the bundles were inserted downwards for a depth of 15 cm, leaving the label tape out. This operation can be facilitated by engaging the terminal loops of the bundled threads by means of a flat-bladed screwdriver or of an open wrench, and pushing the threads down to the desired level. The helping tool is then slid out from the soil. The spade is then used to "earth-up" the soil sideways by inserting it, again, a few centimeters behind the first cut, parallel to it, and pushing gently towards it to close the open rim and ensure contact between the threads and soil. The bundles were left in place for 7 days, after which they were removed by pulling the tape ends, placed between paper tissues to dry possible humidity, and subsequently used for the breakage test (Figure 2).

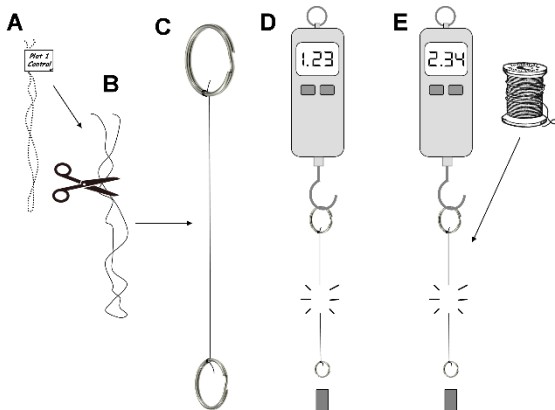

**Figure 2.** Measuring the degradation that occurred during the in-soil incubation. (**A**) The threads are cut from the retrieved bundles; (**B**) the upper portion that protruded outside of the soil during incubation is cut off; (**C**) the useful portion is secured to the keyrings (other hardware can work equally well including Hoffman clamps, Bollard grips, and alike); (**D**) one ring is hung to a dynamometer (or portable weighing scale) hook, and the opposite is pulled steadily until the thread breaks; the recorded peak value is read on the display; (**E**) a corresponding measure, previously taken with the brand new (unburied) thread, is used as a reference comparison.

The weight required to snap the thread (i.e., the weight to apply to it until the thread snaps broken) is related to the thread fragility. The portions of the threads used for the test were cut with scissors from the bundle by discarding the 5 cm top portions from the tape end; each free end generated by the cut was tied to a keyring by pinching the end in and coiling the thread three times around the ring. A digital dynamometer (IMAD ZP, ELIS Electronic Instruments and Systems, Rome, Italy) measured the peak force required to rupture the thread upon applying progressive traction force by pulling its ends. The "peak" function mode was selected in order to read on the display the maximum load withstood by the thread before it broke. In order to refer such value to the extent of increased fragility with respect to the native unexposed (brand new) thread, data were compared to the average values of resistance of the unburied threads of each type. These standards also included the pre-treated but non-buried versions of the N- or P-soaked threads.

The residual resistance to breakage (R) was expressed using the following formula: $R = (R_i/R_{ni}) \times 100$, where R is the resistance percentage; $R_i$ is the rupture value of the thread buried in the soil; and $R_{ni}$ is the rupture value of the new (unburied) filament. In order to express the value as biological activity, the resistance percentage was converted into a degradation percentage (D) by subtracting resistance percentage values from 100. Statistical analysis of the data was performed using CoStat 6.4 (CoHort Software, Birmingham, UK).

## 3. Results and Discussion

### 3.1. Cotton Threads' Reporting Performance

The least degradative microbial activity occurred in the non-fertilized plots (0) with the non-supplemented thread (Figure 3, control cotton, upper panel, white histogram bars), followed by the 0 + r, which was also non-fertilized but received the annual burying of the crop residues. Increasing degradation rates occurred in plots receiving the fertilization treatments (Figure 3). The degradation occurring in parallel on the threads pre-treated with N or P and incubated in the same plots can be directly compared with that of the non-treated thread by vertically inspecting the panels in Figure 3.

The plain thread had the lowest value (14%) in the zero-fertilization plot (first plot on the left, 0) whereas the treated threads showed an increase of degradability, reaching values above 50% (thread + N) and 60% (thread + P), while in the fertilized plots this stimulation was smaller, arguably since their

microbial populations were more satisfied in their N and P requirements. Such differences indicate the capability of the tool to serve as a proxy or index of soil N and P availability.

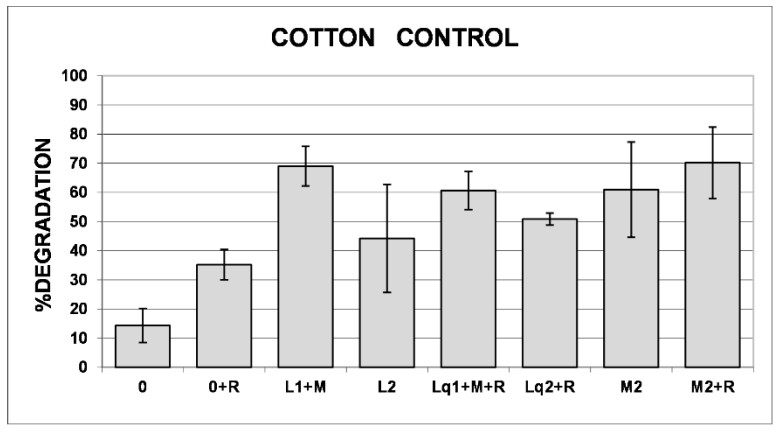

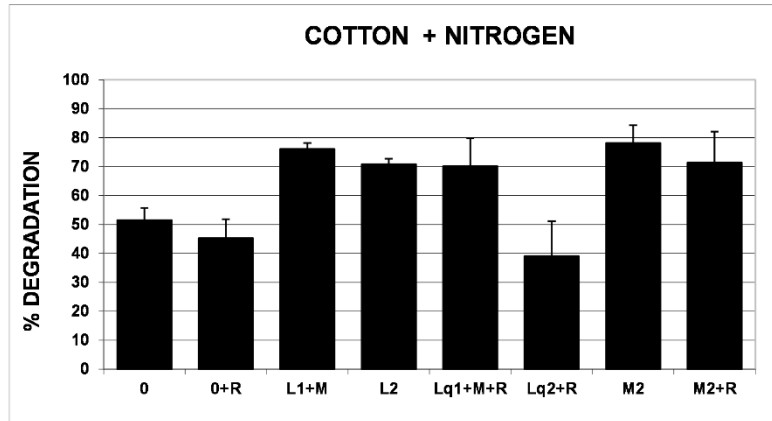

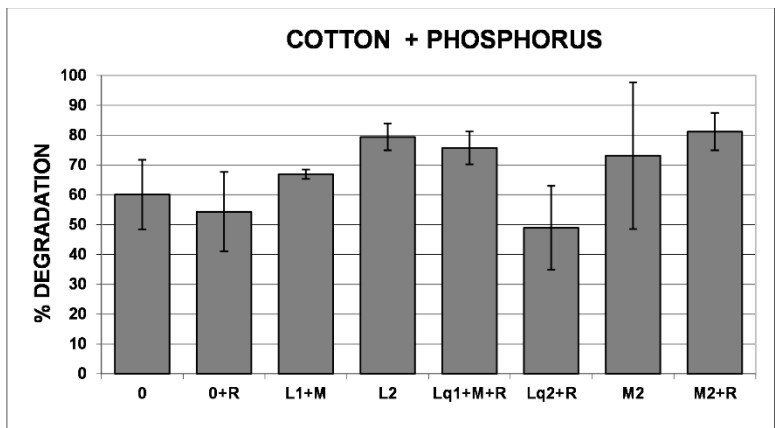

**Figure 3.** Percent of degradation, with respect to the unburied standard cotton thread, observed in the differently fertilized plots of the corn-cropped field during the seven-day incubation. Top panel: untreated thread; middle: thread pre-treated with N solution; bottom: thread pre-treated with P solution. The means of three replicates (independent field plots in the randomized block design) are shown. Bars report the standard deviation.

While the top panel (control cotton thread) had correctly spotted the plots receiving low fertilization inputs, the comparison with the treated ones is an estimate of potential nutrient deficiency. The gap between the degradation recorded with the untreated thread and that attained by the treated one is the difference that allows for the estimation of the severity of the specific lack. In addition,

such delta-value prompts a farmer with the suggestion of the needed proportion of fertilizer to correct the situation.

An ever-growing knowledge on plant-microbe interactions concurs on the fact that microbes are, to a large extent, living mediators that facilitate carbon oxidation and mineralization of organic forms of N and P to inorganic, bioavailable forms. The threads response in absolute terms rests on the same principle of the cotton strip assay [7,9,12] and depends on the chemical and physical variables (e.g., pH, moisture, temperature, texture) ruling plant and microbial physiology. Degradation values in a given soil are; therefore, changing along with seasonal events. Nevertheless, the relative difference in the degradation of plain vs. N- or P-supplemented threads provides a built-in control that enables the assessment of nutrient deficiencies in spite of the fluctuation of the overall parameters. Another element worth noting is the overall shape of the graph in the two nutrient-primed panels (N, P), in which the histogram series shows a consistent trend of the corresponding histogram heights. A consistently lower mean of the plots 0, 0 + r, and of the one receiving the double dosage of slurry (Lq2 + r), against the high values of the others are noticeable.

The extent of degradation on the cotton fiber can, thus, be taken as a measurable proxy reporting on the promptness of cellulolysis, which in turn is entailing the turnover of mineral nutrients from organic matter. The method capitalizes the principle of the cotton strip assay [7–9] but goes beyond each of their subsequent elaborations [10–12], as it introduces the nutrient-induced degradative acceleration. In this respect the N- or P-spiked versions of the threads can provide the information on whether the soil needs to be supplemented or not to achieve optimal productivity, allowing the farmer to timely plan a corresponding correction. Conversely, if no gap is shown by the comparison, it could be possible to avoid an unnecessary treatment and benefit from saving fertilizers, added costs of oil, and manpower, sparing these impacts to the environment.

On the nature of the differential predictive capability of the treated vs. untreated threads, the underlying mechanism is based on the fact that, besides the C supplied by the threads, soil microbes require also N and P in order to grow. Availability of those, either in the soil or on the threads, enables their exponential doubling to occur, causing the increased degradation rates.

*3.2. Silk Threads' Reporting Performance*

Regarding the silk fiber, it represents, instead, a component from the animal world and reports the proteolytic enzymatic potential. Organic matter of animal origin is quantitatively much less abundant as an input into soil and the significance of a protein thread as a reporter is rather different from that of the cellulose thread. Nevertheless, it offers a complementary perspective in terms of soil enzymology and can reveal different details, as shown in Figure 4.

The overall level of degradation during the one-week incubation was less intense than that of cotton, and no peaking differences occurred across the differently fertilized soil plots. The N-spiked silk threads did not show increments in degradation either. This is in part expected; being that silk is made of proteins (sericine and fibroine), thus the plain thread itself already contains 16.5% N as amino acids. Presumably, due to such nature, the pretreatment of the thread with $(NH_4)NO_3$ did not stimulate silk degradation. On the contrary, P pretreatment (bottom panel) increased silk degradation from 20% to 23–36%. These increases confirm the P-limiting conditions of the soils in Legnaro (Giardini et al., 2004). Another positive aspect of silk is the higher reproducibility of measures, as shown by the more contained standard deviation bars (Figure 4) when compared with the cotton results (Figure 3). Likely the high strength of the silk fiber also accounts for its sharper breakage and uniformity of results compared to the cotton threads. The cotton was more labile than silk, showing peaks of degradation as high as 88%, in some of the highly fertilized plots, which was more than double than the maximum observed for silk. Cotton also displayed a higher sensitivity of resolution, as its minimum of degradation was lower than 10% in non-fertilized parcels. On the other hand, silk, as mentioned above, performed better in terms of stability and homogeneity of results, showing sharper reproducibility of the data. The fact that silk contains nitrogen; nevertheless, does not

automatically imply that its pre-treatment with mineral nitrogen would be useless; silk nitrogen is in the form of amino acids and thus an access to it requires proteolytic proficiency, but the soil microbial guilds involved could also benefit from a priming conferred by promptly soluble mineral nitrogen sources, such as ammonia or nitrates. This justifies the maintenance of the N-treatment also in the silk fibers.

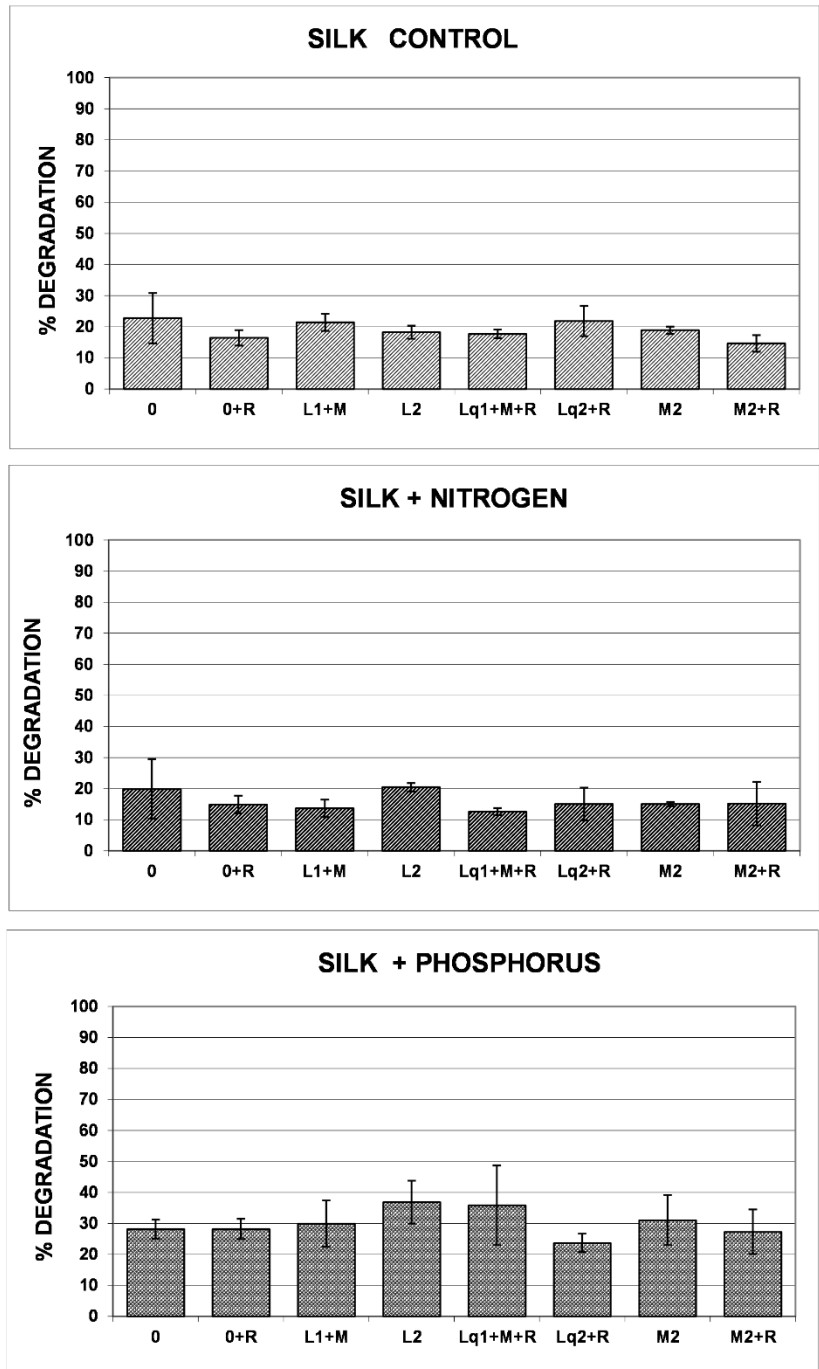

**Figure 4.** Percent of degradation, with respect to the unburied standard silk thread, observed in the differently fertilized plots of the corn-cropped field during the seven-day incubation. Top panel: untreated thread; middle: thread pre-treated with N solution; bottom: thread pre-treated with P solution. The means of three replicates (independent field plots in the randomized block design) are shown. Bars report the standard deviation.

### 3.3. Productivity Prediction by the Fertimetro

The final aspect that we verified was the correspondence between the degradation values observed and the actual fertility of each plot, measured at crop harvest, both as grain yield (quintals per hectare) and as straw (corn stover biomass of stalks and leaves). These values, obtained in October after harvesting, were compared with those that had been recorded by the threads in late May. The data from the respective field units are shown in Figure 5, where plots are arranged in decreasing order of productivity, left to right. The series of thread degradation data is consistently matching that of the yield data obtained five months later. The straw series (that takes into account the major part of the plant biomass and not the sole seeds) is, in this respect, remarkably faithful to the corresponding thread degradation values. The corresponding regression curves are shown on the right in the same figure.

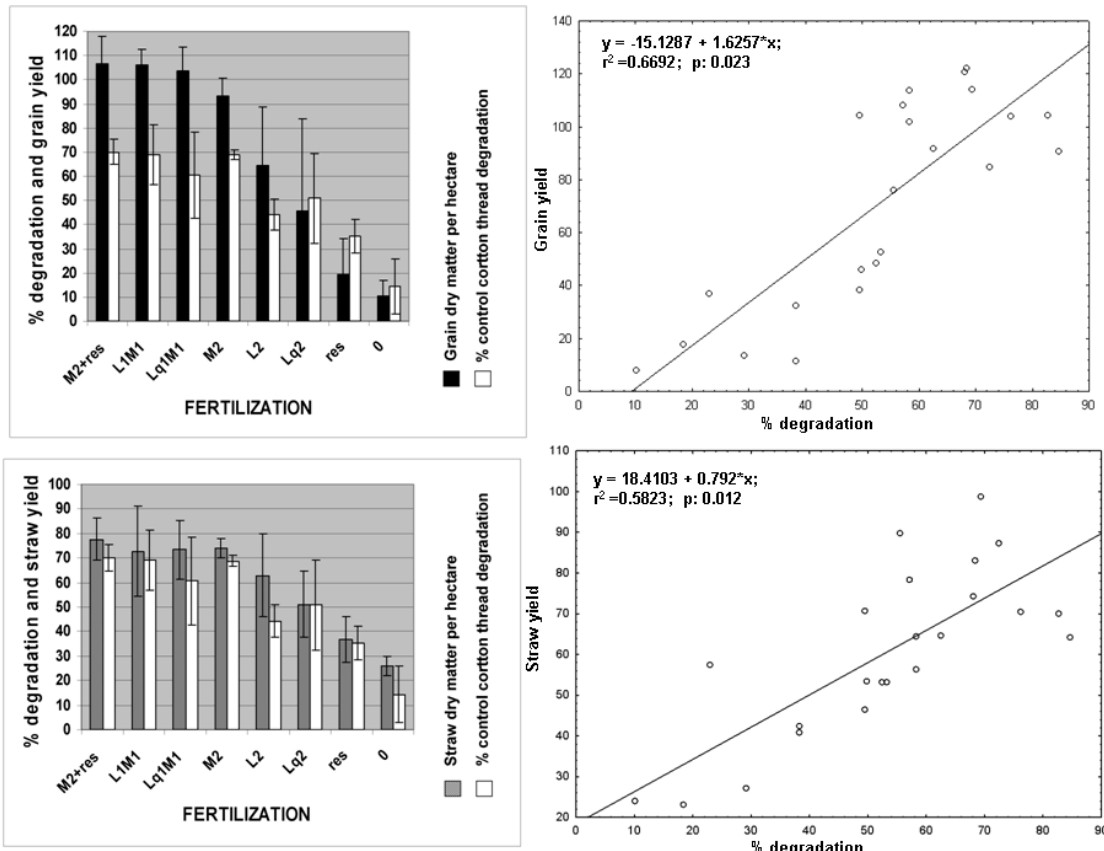

**Figure 5.** Pairwise comparisons of the dry matter yield (quintals per hectare) of corn at harvest (October) with the corresponding values of the Fertimetro control cotton thread degradation recorded in May, in the differently fertilized plots, arranged, from left to right, in order of decreasing yield. The means of three replicates (independent field plots in the randomized block design) are shown. Bars report the standard deviation. Top panel: comparisons for grain yield; bottom panel: comparisons for straw yield; left panel: means ± SD; right panel: regression analysis between the same data series. Crop yield data are in quintals.

The whole dataset relative to the crop yield and other nutritional parameters of the grain, versus that of the thread degradation values, were used to check for Pearson r correlation, and the results of the statistically significant correlations are shown in Table 2. All the three types of cotton threads showed very strong correlations with grain yield; the plain cotton thread (control) was highly correlated with straw yield (data also shown in the bottom panel of Figure 5). The non-spiked control thread turned out to be also correlated with some compositional parameters of the corn grain. In particular, negative correlations with fat content and humidity were supported by the *p*-values. The reason by which

the correlations between the plant data and the degradation values of the N- or P-supplemented threads were slightly lower can be interpreted: The untreated threads record the actual field situation experienced by the microbial communities, for example, in the unfertilized plots, the lowest values of crop yield and the lowest values of degradation occur, but in the same plot, the N-or P-spiked threads receive an on-thread-only compensation, which helps the degrading microbes but not the plants. For this reason, the response of the untreated threads can be assumed to be the one which is truly mirroring the plants' status in that field, which justifies the highest correlation scores.

**Table 2.** Pearson coefficients (r) of statistically significant correlations between corn harvest data and the corresponding values of the Fertimetro threads' degradation recorded five months earlier. *p*-value coding: *: $p < 0.05$; **: $p < 0.01$; ***: $p < 0.001$.

| Corn Yield (October) | Fertimetro Data (May) | r Coefficient | *p*-Value |
|---|---|---|---|
| Yield of corn grain (tons per hectare) | % degradation cotton control | 0.78 | 0.0001 *** |
| Yield of corn grain (tons per hectare) | % degradation cotton + nitrogen | 0.70 | 0.0001 *** |
| Yield of corn grain (tons per hectare) | % degradation cotton + phosphorus | 0.55 | 0.0049 ** |
| Yield of corn straw (tons per hectare) | % degradation cotton control | 0.73 | 0.0001 *** |
| Fat content % in corn grain | % degradation cotton control | −0.63 | 0.0009 *** |
| Starch content % in corn grain | % degradation cotton control | 0.47 | 0.0200 * |
| % humidity in corn grain | % degradation cotton control | −0.63 | 0.0009 *** |
| Protein content % in corn grain | % degradation silk + phosphorus | −0.43 | 0.0339 * |
| Starch content % in corn grain | % degradation silk + phosphorus | 0.42 | 0.0395 * |

The silk thread spiked with P revealed also some less strong, but still significant (*p*-value < 0.05), correlations with the protein and starch content, respectively, of negative and positive sign. It is worth noting that, besides crop yield, other aspects of the product's biochemical composition showed correlation with the thread degradation levels. It is not clear whether some underlying physiological phenomena could explain these interesting additional findings, which deserve further experimentation. At this stage; however, it can be commented that the relationships between underground communities are complex and articulate; it could be speculated that the network of microbe–plant interplay of the soil ecosystem could contribute to a cascade of phenomena that could ultimately affect, also, the nutritional value of the crop products.

The field trials demonstrated the suitability of the method to probe the level of fertility of soils, and to point out the proportion of nutrient requirements that would be needed to achieve the maximum level of crop productivity. Besides the major macronutrients for agricultural fertilization, the principle may be tested to detect oligoelements deficiencies.

In comparison to soil laboratory analyses, the present methodology introduces potential shortcuts. With traditional analyses the user has to send the soil sample to a laboratory, from which will later receive a list of values that need to be interpreted to formulate hypotheses on potential soil fertility. The principle hereby shown, instead, can be self-performed in one's own farm and is based on an in vivo test, yielding a single parameter, relative to thread degradation percent, which appears statistically correlated with plant productivity.

In conclusion, the key features of the method are: (1) The possible increase of productivity per hectare due to the correct fertilizer rate; (2) foretelling harvest yield several months in advance (in time to still optimize it when needed); (3) money savings, avoiding superfluous fertilizer waste and environmental pollution from excess fertilizer runoff; (4) easily interpretable data, which can be turned into exact specific amount of added fertilization (decisional charts or online suite for different crops and climates can be planned); (5) self-made analysis (km 0), at low cost, high response speed, and the possibility to repeat tests for an unlimited number of times. The method has also been automatized with a technical modification [16], in which threads are hooked in a frame with springs that keep them already under a tension equal to half of the strength that would break them. When the residual resistance is eroded by the microbial degradation, the thread snaps (in-field) and the corresponding spring's stick hits a battery-operated data logger, recording the timing of the breaking

event. Wireless/bluetooth data collection (also via flying drones) is, in this case, possible. Such that, in a way, the variable that is measured becomes the time instead of the weight; the more active the microorganisms are on that thread, the earlier it will break. The extent of anticipation compared to the untreated thread expresses the deficit of that nutrient compared to its optimal values.

**Author Contributions:** Conceptualization, G.C., S.T. and A.S.; formal analysis, G.C., P.T. and A.S.; funding acquisition, G.C. and A.S.; investigation, G.C., P.S., F.M., A.B., R.P., M.B. and A.S.; methodology, G.C., S.T. and A.S.; project administration, A.S.; supervision, A.S.; validation, A.S.; writing—original draft, A.S.; writing—review and editing, A.S.

**Funding:** The research was funded by PRAT 2015, code: CPDA154841, (Progetti di Ricerca di Ateneo, University of Padova; "Fertimeter: a method, device, and project to reduce external inputs in agriculture and optimize land productivity"). The work was also supported, in part, by funds obtained as the winner of the second prize for best technological improvements in the national competition "Proof of Concept Network" (PoCN), issued by Area Science Park Consortium, Trieste.

**Acknowledgments:** The authors thank Cristina Benetti for assistance in the field trials and dynamometer readings. The license of the patent for different countries is available from the University of Padova for negotiations by investors interested in serial production of the instrument, development of the technology, and product marketing.

**Conflicts of Interest:** The authors declare no conflicts of interest.

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
