# Peer review of "Fertimetro, a Principle and Device to Measure Soil Nutrient Availability for Plants by Microbial Degradation Rates on Differently-Spiked Buried Threads"

_soilsystems, doi:10.3390/soilsystems3010003_

Round 1

Reviewer 1 Report

Review report for manuscript entitled: “Fertimetro, A Principle and Device to Measure Nutrient Availability for Plants by Microbial Degradation Rates on Differentially–Spiked Buried Threads” submitted to Soil Systems

Summary

This study presents a novel method to estimate soil productivity for maize growth by employing an in situ, nutrient-impregnated fiber (cotton or silk) degradation test to assess soil microbial decomposition potential as a proxy for mineralization/nutrient release. The difference between thread fragility between N or P-impregnated threads versus un-impregnated threads (control) serves as the quantitative measure of nutrient availability. In this study, the test was applied to long-term maize fertility plots at the University of Padua, Italy and subsequently calibrated with a maize yield trial.

Broad Comments

The study described herein is well designed and evaluates a novel method to gauge potential crop yield on soils of varying properties. The method is inexpensive, easy to use and appears well correlated with differences in maize yield. While the method presented appears to offer potential, substantial revision is needed before publication can be considered. In general, the results and discussion needs to be revised to be more concise and focus on how the authors’ findings relate to other studies. Right now the discussion is hard to follow and disjointed in several parts. Please refer to specific comments and suggestions that follow.

Specific Comments

Abstract

Line 20. What is meant by “plain” version?

Line 25. Consider stating the R2 value for the linear regression and P-value.

Line 26. Suggest changing “correct” fertilizer rates to “improved” fertilizer rates.

Line 28. Revise “are worth of investigation” to something like “…are important considerations”

Introduction

Line 35. Replace ‘has’ with ‘have’ after ‘variety of methods’.

Line 41. extra space after ‘require..’

Line 45. replace ‘cheapness’ with “cost savings or ‘cost-effectiveness’

Line 46. Suggest changing ‘aptitude’ with ‘potential’

Line 71. ‘…2 kg’ of what? Force?

Line 71-72. Change to: ‘This enables testing of residual…’

Line 80-84. Change to: “This allows measurement of both (no underline)…delete “that” after ‘both’. Please clarify what is meant by N and P-inducible’? Assuming this implies mineralization of organic matter and release of plant-available N and P?

Line 87. ‘..the manual’. Do you mean ‘method’?

Materials and Methods

Line 96. What about inorganic fertilizers? What rates of organic fertilizers were applied at and what is the P application rate?

Line 103. What method/extraction procedure was used for phosphorus? Does this represent total soil P or is it an estimate of labile P?

Table 1. “Fertilization routine…’ replace ‘routine’ with ‘regime’ or ‘treatment’

How were maize yields determined?

Please clarify the difference between ‘slurry’ versus ‘manure’. What was the solids content of each and P content? How much P was applied manure applications?

Line 139. Insert a comma after ‘exposure’

Fig. 2 caption. Extra period at end.

Line 171. Replace ‘elaboration’ with ‘analysis’

Consider stating the experimental design and statistical model used for the trial. Also consider explicitly stating your hypothesis.

Results and Discussion

Line 175. First sentence seems like it should go into the Materials and Methods section.

Line 178: What is meant by “cultural residues”? Does this mean crop residue?

Line 187: Suggest changing ‘…of the tool to record plant nutrient…’ to serve as a proxy or index of soil N and P availability…’

Figure 3. Suggest omitting all gray backgrounds on figure panels as it makes some of the bar graphs difficult to read. Please define various fertility treatments as indicated by the different letter/number combinations of the treatments.

Line 196. What does ‘cross-verification’ mean? Is this the inclusion of the ‘control’ or untreated thread at each site?

Line 197. Suggest modifying to something like:  ‘… treated one is an estimate of potential nutrient deficiency…’

Line 202. Suggest revising to: ‘too a large extent living mediators that facilitate carbon oxidation and mineralization of organic forms of N and P to inorganic, bioavailable forms’.

Lines 202-210. Can this section be condensed to a few main points with references to other relevant studies? How do your findings help fill knowledge gaps with respect to organic matter decomposition, nutrient release, and maize yield potential?

Lines 211-214. There is no need for a separate paragraph. Consider integrating this section into a revised discussion of the above factors (e.g., soil/seasonal effects on potential decomposition)

Line 212. ‘..parallel trend’. Consider replacing ‘parallel’ with consistent.

Lines 215 to 231. This whole section needs to be rewritten to focus on your main findings in relation to other studies. Other relevant studies from the literature must be discussed here. How do your findings help better predict N and P release from organic carbon turnover? What have other studies shown in relation to your findings? How can your results be used to help improve estimates of N and P requirements of maize?

Fig. 4. Omit gray background on panels- white background will show differences better.

Line 249. Insert a comma after ‘contrary’

Line 249. None of the means appear to reach 40%. Revise this to reflect the actual range.

Line 250. Increases is misspelled

Line 255. Why a new paragraph here?

Lines 250-259. How useful are the silk results if N mineralization potential is not assessed and only P? What have other studies found relative to your findings?

Lines 261 to 270. This section should be condensed down to a few main points and supported by literature. What units are the yield data presented in?  Is this relative yield or tons/ha?

Figure 5. Figures need to be enlarged and are not readable. Regression equations are illegible. What was the P-value for the overall regression? What does ‘synoptic’ here mean in caption?

Lines 279-287. This section needs to be rewritten to focus on possible physiological mechanisms that may help explain these findings. Have other studies found similar correlations? Are these biologically significant or just statistically?

Table 2. The best correlation was for untreated control cotton thread. Does this imply N and P-spiked threads affected microbial dynamics and reduced decomposition? Your discussion should address this observation and possible explanations.

Line 293. What is meant by ‘top fertility’?

Lines 291-315. This whole section needs to be revised to be more concise and focus on your most relevant findings and how your results push the science forward.  Can your results be used to better predict maize yield potential, N and P removal and thus fertilizer rates?

What are the limitations of your method? Only one season’s worth of yield data were collected. Is this sufficient to fine-tune N and P rates across different soils in the region? How would changes in weather/soil moisture affect your test results? What field conditions are recommended for optimizing test results? How does this test compare to a standard agronomic soil test currently used to make fertilizer recommendati

Author Response

A point-by-point series of answers is attached as docx file

Reviewer 2 Report

-The results presented in this manuscript is only for one type of soil. Did authors use different types of soils with varied physico-chemical properties?  if so, it would be proper to include such data in the manuscript.

In Figures 3 and 4 - since the X and Y axis are the same, the plotting of data for control, N+ control and P+ control can be combined in one graph to better visualize the comparison between the control and treatments ( the statistical analysis should reflect this comparison and shown in graph).

Author Response

A point by point series of answers is attached as docx file

Round 2

Reviewer 2 Report

none